# pH Controlled Nanostructure and Optical Properties of ZnO and Al-Doped ZnO Nanorod Arrays Grown by Microwave-Assisted Hydrothermal Method

**DOI:** 10.3390/nano12213735

**Published:** 2022-10-24

**Authors:** Lamia Al-Farsi, Tewfik M. Souier, Muna Al-Hinai, Myo T. Z. Myint, Htet H. Kyaw, Hisham M. Widatallah, Mohammed Al-Abri

**Affiliations:** 1Department of Physics, College of Science, Sultan Qaboos University, Muscat P.O. Box 36, Oman; 2Department of Process Engineering, International Maritime College Oman, Sohar P.O. Box 532, Oman; 3Nanotechnology Research Center, Sultan Qaboos University, Muscat P.O. Box 17, Oman; 4Department of Petroleum and Chemical Engineering, College of Engineering, Sultan Qaboos University, Muscat P.O. Box 33, Oman

**Keywords:** Al-doped zinc oxide, semiconductor, crystal growth mechanism, optical properties, photoluminescence, nanostructure, microwave-hydrothermal deposition

## Abstract

The low-temperature microwave-assisted hydrothermal method was used to successfully grow pure and Al-doped ZnO (AZO) nanorod (NR) arrays on glass substrates. The combined effects of doping and pH on the structural properties, surface chemistry, and optical properties of all samples were investigated. Thermodynamic-based simulations of the growth solution were performed and a growth mechanism, that considers the effects of both the pH and Al-doping, is proposed, and discussed. Tuning the solution pH is key parameter to grow well-aligned, single crystal, highly packed, and high aspect ratio nanorod arrays. Moreover, the optical absorption in the visible range is enhanced by controlling the pH value. The PL spectra reveal a shift of the main radiative emission from the band-to-band into a transition involving deep defect levels of Zinc interstitial Zn_i_. This shift is caused by an enhancement of the non-radiative components (phonon relaxation) at high pH values. The production of well-ordered ZnO and AZO nanorod arrays with visible-active absorption/emission centers would increase their potential use in various applications.

## 1. Introduction

ZnO is recognized as a multifunctional material due to its unique physical and chemical properties. It also possesses the richest family of nanostructures among all materials; this includes nanocombs, nanorings, nanocages, nanosaws, nanospirals, nanosprings, nanobelts, nanowires, and nanorods [1,2,3,4,5,6,7,8,9,10,11,12,13]. Among these, 1D ZnO NR arrays have drawn more attention due to their potential use as building blocks for a wide variety of nanodevices, including nanosensors, UV photodetectors, solar cells, light-emitting diodes, nanogenerators, field-effect transistors, and photocatalysts for the degradation of dyes and pollutants [10,11,12,13,14,15,16,17,18,19,20,21]. However, their morphological, electrical, and optical properties must be tuned to optimize their use in these applications.

ZnO belongs to the II–VI semiconductor family with a wide bandgap of 3.37 eV and a large exciton binding energy of 60 meV at room temperature. Usually, it grows into an n-type semiconductor [22]. It was widely admitted that the n-type conductivity originates from the large concentration of zinc interstitials and oxygen vacancies acting as electron donors to the conduction band [22,23]. However, density functional theory (DFT) calculations have shown that the “unintentional” residual hydrogen impurities that create shallow donor levels, are responsible for the n-type conductivity of ZnO.

It is crucial to control the n-type doping of ZnO NRs to tune their physical properties. This is achieved by trivalent extrinsic dopant atoms such as Al, Ga, and In, acting as donors when substituting for Zn sites in the ZnO lattice. Extrinsic doping proved to play a crucial role in improving the properties of ZnO NRs [24,25,26,27,28]. Among these, Al dopants can reach very high n-type conductivity without deterioration in optical transmittance [29,30,31], and therefore lend themselves to be ideal candidates for such use. In previous works [32,33], we have shown that Al-doping increases the photocatalytic activity of the ZnO NR array under visible light irradiation.

There have been multiple attempts to synthesize vertically aligned Ga-doped ZnO NRs [27,34] and AZO NR arrays [31,32,35] on different substrates. Out of the techniques employed, a hydrothermal method (or chemical bath deposition) appears advantageous owing to its low cost and low growth temperature [9,30]. This method, proven effective in growing well-aligned ZnO NR arrays, has been extensively studied and optimized for various parameters including the temperature, the quality of the seed layer, the solution concentration, and pH. However, achieving perfectly aligned ZnO NRs doped with group III ions turn out to be challenging. The difficulty arises when high doping levels, required for improving optoelectronic properties, are sought. Fuchs et al. have successively grown dense doped ZnO NRs array using aluminum nitrates, showing that at high doping concentration, typically 10 mM, the NRs alignment is completely deteriorated [36]. Qi Yu et al. have found that adding Al^3+^ dopants gradually changes the nanostructure of ZnO, from nanorods into nanosheets [37]. Our previous work reported an enhancement of NRs growth efficiency by using a microwave-assisted hydrothermal deposition technique. Compared to standard heating, the use of the microwave offers fast volumetric heating, shorter reaction time, and the rapid and homogenous nucleation of ZnO crystals [12]. Using this technique, well-aligned and long ZnO NR arrays were obtained. However, Al dopants additions were found to deteriorate the alignment and the crystallinity of ZnO NR arrays [32]. Similar observations have been reported for Ga-doped ZnO NRs [38,39]. The vertical alignment of doped ZnO NR array, leading to an increase in surface area, plays an important role in their suitability for wider use in practical applications. Moreover, instead of mere speculations, it is difficult to conclude the real origin of the reported enhancement in electrical/optical properties of doped ZnO NRs as being the effect of extrinsic doping or induced morphological changes.

The present study aims to optimize the microwave-assisted hydrothermal method to achieve a high aspect ratio, well-aligned, and dense arrays of ZnO and AZO NR arrays on glass substrates. The combined effect of Al-doping and growth solution pH on morphology and structural, compositional, and optical properties of ZnO NRs are investigated. Thermodynamic simulations, of the growth solution, are used to discuss the growth mechanism of ZnO NRs by considering the effect of doping and pH.

## 2. Experimental Details

### 2.1. Materials

Zinc acetate dihydrate [Zn(CH_3_CO_2_)_2_·2H_2_O ≥ 99%] was supplied from BDH Laboratory supplies. Zinc nitrate hexahydrate [Zn(NO_3_)_2_·6H_2_O ≥ 98%] and aluminum nitrate nonahydrate [Al(NO_3_)_3_·9H_2_O ≥ 98%] were obtained from Sigma-Aldrich (St. Louis, MO, USA). Hexamethylenetetramine (HMTA) [C_8_H_12_N_4_ ≥ 99%] and ammonia [(NH_3_) concentration = 25%] were provided by Merck KGaA. Deionized (DI) water used is reagent-grade was produced by AXEON RF-Series Residential Reverse Osmosis Systems.

### 2.2. Synthesis

The substrate used for growing ZNO NRs was soda-lime glass from SPI supplies. All glass substrates were subjected to the following cleaning procedure through 20 min cleaning with soap and DI water, followed by another 20 min cleaning with ethanol and acetone, and finally drying in an oven at 90 °C. The cleaning steps were performed in the ultrasonic bath. ZnO seedlayer was formed using a spray pyrolysis system by spraying 10 mM of zinc acetate solution on heated glass substrates at 350 °C. The ZnO NR arrays were grown using microwave-assisted hydrothermal method on ZnO seed layer. 60 mM of HMTA was mixed with 60 mM of Zinc nitrate (1:1 ratio) to have a final precursor’s concentration of 30 mM. The pH of the growth solution without ammonia is close to neutral while in alkaline solution it was adjusted using ammonia additions (pH = 9.5 to 11). The seeded glass substrates were placed in a petri dish with 200 ml of the precursor solution, and put in a microwave oven under a power of 180 W for 45 min. Under these conditions, the temperature of the growth solution was observed at ~95 °C to be close to the boiling point of water. The microwave oven used is a 2.45 GHz Samsung MC28H5015AW. After cooling down to room temperature, the growth solution was replaced by a fresh one and the process was repeated for extra four cycles. The pH of the growth solutions was measured, using the Seven Compact S220 pH meter from Mettler Toledo, at the beginning (pH_i_) and the end (pH_f_) of each cycle. Finally, the slides were cleaned with DI water and put in a furnace at 90 °C for 10 min. The AZO NRs were prepared as in the previous steps except that aluminum nitrate nonahydrate was added to the zinc nitrate hexahydrate precursor (Al/Zn molar ratio = 5 mol%) while maintaining the precursors’ concentration at 30 mM. The Al/Zn ratio represents the expected Al-doping level, but the effective doping level will be quantified using XPS and TEM analysis. Table 1 below lists the labels of the studied samples along with the experimental growth conditions. 

### 2.3. Characterization

The crystalline structure of ZnO and AZO NR arrays was characterized by X-ray diffraction (XRD) using a Rigaku X-ray diffractometer operating at 40 kV and with Cu K*a* radiation (*λ* = 0.1542 nm). The PDXL software was used to fit the XRD diffraction patterns and compute the crystalline parameters: crystallite size *D*, lattice parameters *a* and *c*, and Zn-O bond length using a known procedure [32,33]. The bond length was calculated using the equation:$$L=\sqrt{\frac{a^{2}}{3}+{\left(\frac{1}{2}-u\right)}^{2}c^{2}}\;\mathrm{with}\;u=\frac{a^{2}}{3c^{2}}+\frac{1}{4}$$

The NRs array morphology was examined using high-resolution field emission scanning electron microscopy (HR-FESEM, Joel SEM JSM-7800F). To confirm the doping efficiency and the surface chemistry, further investigations were carried out using X-ray photoelectron spectroscopy (XPS) by a multi-probe X-ray photoelectron spectroscope from Scienta Omicron. The XPS spectra were analyzed using the CasaXPS software. High-resolution transmission electron microscope (HRTEM) by a JEOL (JEM-2100F), operating at an accelerating potential of 200 kV and equipped with energy-dispersive X-ray spectrometer (EDS, Joel EX-24063), was used to perform elemental mapping on AZO NRs and the corresponding EDS spectra. Optical absorption spectra of all samples were recorded using a Uv-Vis spectrometer (Lambda 12 PerkinElmer) in the 300–800 nm spectra region. Room temperature photoluminescence (PL) spectra were carried out using PerkinElmer LS55 fluorescence spectrophotometer under excitation wavelength of 325 nm. 

### 2.4. Thermodynamic Simulations

To investigate the pH effect on the growth mechanism, the solution of growth is simulated using thermodynamic calculation under the Visual MINTEQ software (version 3.1). The ‘model’ solution consists of water, hydronium, hydroxide ions, ammonia and ammonium ions, and zinc and aluminum species. The Zn(II) species are Zn^2+^ ions and Zn(II) hydroxides and amine complexes, while Al(III) species are Al^3+^ ions and Al(III) hydroxides. The equilibrium calculation was run at 95 °C (growth temperature), with initial experimental concentrations of the chemicals. The speciation diagram of all species is obtained by sweeping the pH from 4 to 12. 

## 3. Results and Discussion

### 3.1. Morphological and Structural Properties

Figure 1a shows XRD patterns for undoped ZnO NRs. All NR arrays crystallize within the hexagonal wurtzite crystalline structure in agreement with crystallography open database COD 1011259 of the zincite ZnO phase [40]. The samples exhibit diffraction peaks of (100), (002), (101), (102), (103), and (104) crystalline planes, but with a very strong preferential orientation along (002) reflection. This suggests that each nanorod can be regarded as a single crystal that grows along the *c*-axis of the wurtzite crystal structure. Similar results are observed in AZO NRs samples with the aforementioned diffraction peaks and preferential (002) orientation (Figure 1b). However, Al-doping is found to induce new XRD peaks associated with (110), (112), and (201) crystalline planes. Interestingly, a low angle diffraction peak at about 2θ~30.8° is also evidenced in some ZnO and AZO samples; mainly those grown in alkaline media (pH = 10.5 or 11). A similar peak is observed by some authors and is attributed to zinc hydroxide Zn(OH)_2_ phase [41]. It is important to emphasize that the growth of crystalline ZnO is a two-step reaction, and the existence of zinc hydroxide could be explained by an incomplete oxidation process (as shown in reaction 1). It should be noted that Al secondary phases, such as Al_2_O_3_ or Al(OH)_3_, have not been observed. If such phases exist, they should be in low quantities below the XRD detection limit. In order to rule out this possibility, the quantification of the Al content was performed using XPS and TEM-EDS analysis.
(1)$$2OH^{-}+Zn^{2+}\;\to Zn{\left(OH\right)}_{2}\;\to ZnO+H_{2}O\;$$

Besides the above description, the main finding is that the intensity of the (002) peak is highly dependent on the pH of the precursor’s solution and Al-doping. In both ZnO and AZO samples, the highest (002) intensity was recorded at pH = 11, whereas the lowest intensity is recorded at pH = 9.5. Moreover, the intensity of the main XRD peak decreases upon Al-doping at all pH values. The XRD intensity could be related to the length of the NRs, suggesting longer ZnO NRs in comparison with AZO NRs at a given pH.

The XRD patterns were analyzed using the PDXL software to compute various crystalline parameters. Each XRD peak was fitted using the pseudo-Voigt function, which is a linear combination of Gaussian and Lorentzian components. The average crystallite size *D* was determined from the main (002) XRD peaks. The results (Table 2) show that the lattice parameters, the *c*/*a* ratio, and the Zn-O bond lengths, of ZnO and AZO samples, are in good agreement with those reported in the literature [42,43]. This shows that the Al is well incorporated in ZnO wurtzite lattice suggesting efficient doping. Upon Al-doping, the peaks corresponding to (002) plan showed a slight shift towards a lower angle which could be due to the progressive incorporation of Al^3+^ ions into the lattice of the ZnO.

Figure 2a–d show the top-view SEM images for the undoped ZnO samples grown at different pH values. All samples exhibit dense arrays of aligned nanorods growing perpendicularly to the substrate. However, the morphology and shape of nanorods are highly dependent on the growth solution pH. Under neutral conditions (hereafter referred as ZnO I), the nanorods exhibit a hexagonal-shaped facet, (001) *a*-plane, with a diameter ~65 nm and density of 37 µm^−2^, similar morphology has been reported in the literature [44,45]. On the contrary, needle-like, or pen-like NRs, are observed under the basic condition of pH = 11 (hereafter refereed to ZnO IV). This could be related to a pH etching process of the nanorod’s tip. Our findings corroborate previously reported data [44,46]. Under a basic condition of pH = 10.5, both hexagonal-shaped and pen-like (002) facets nanorods are observed. The diameter of rods is found to increase from 68 nm to 100 nm with a decreasing density from 37 µm^−2^ to less than 20 µm^−2^ for ZnO grown under pH 10.5 and 11, respectively. Figure 2e–f show the cross-section of the ZnO I and ZnO IV samples revealing a high degree of alignment of the NR arrays. Under neutral conditions, ZnO I array has a thickness of 5.2 µm corresponding to an aspect ratio of ~80 whereas, under basic conditions, ZnO IV (pH = 11) array presents a thickness of ~11.4 µm and an aspect ratio of ~114. This corroborates the comparatively high XRD intensity of (002) reflection in ZnO IV sample. We point out that the thickness of ZnO IV sample is one of the highest reported in the literature for hydrothermally grown ZnO nanorods. This is due to the fast kinetics growth under alkaline conditions, the use of the microwave, and the renewal of the precursor solution five times during the growth.

Under a pH value of 9.5, the sample (hereafter referred to as ZnO II) shows a denser array of rods with spherically shaped tips that suggest a thin film columnar structure. These results agree partially with what was reported by Jang et al. [47] where a coalescence of ZnO nanostructure is observed under pH values in the range 9–9.5, but they obtained a budding flower-shaped structure. This difference could be related to the authors’ use of zinc acetates as precursor for NRs growth. Current results suggest that the growth of a dense ZnO array is possible under pH 9.5. Still, the array’s thickness is comparatively small, leading to the formation of low aspect ratio nanorods.

Figure 3a–g shows the top view and the cross-sectional SEM images for doped AZO samples at various pH. SEM images depict AZO I array as large diameter (0.57 μm) flower-like shaped NRs with hexagonal top-facet. These NRs are randomly oriented and only a fraction grows perpendicular to the substrate. Accordingly, under these conditions and unlike the ZnO counterpart, there is no alignment of the NR array. By adding aluminum nitrates precursors, the pH of the growth solution decreases to about 5.5, and a more acidic pH value is expected by increasing the temperature to 90 °C in the microwave. Therefore, the ZnO nanostructures in AZO-I samples are grown in more acidic media compared to ZnO-I samples. Our results are in line with those reported in the literature where a comparable nanostructure is obtained by growing ZnO NRs in acidic media [44,48,49].

Under basic pH values of 9.5 and 10.5, AZO grows as a thin film with dense columnar nanostructure and thickness of 0.21 and 0.75 µm, respectively. The obtained nanostructure is quite similar to ZnO II sample. A low aspect ratio of 4–14 is found when considering the estimated AZO NR diameter of 55 nm.

The top view- and cross-section images (Figure 3d,g) show a 3.1 µm thick array consisting of pen-terminated NRs with an average NR diameter of 55 nm and an aspect ratio of 56. Only at a higher pH_i_ value of 11, a well-aligned and dense (45 µm^−2^) AZO array is obtained. We conclude that the growth of a dense and well-aligned AZO array is only possible under a high basic pH value of 11.

To confirm the Al-doping of ZnO NRs, TEM images with elemental mapping and EDS spectra were recorded on AZO I and AZO IV samples as shown in Figure 4a,b. The AZO NRs were simply dispersed in ethanol and deposited on Cu TEM grid for analysis. The AZO NRs formed bundles when dispersed in ethanol. The EDS-TEM elemental maps clearly show the Al (dark green), Zn (light green), and O (red) spatial distributions. Images revealed a homogenous distribution of elements over ZnO NRs in both samples, which confirms an effective Al incorporation within the bulk and at the surface of the nanorods. The atomic percentage (at. %) extracted from EDS spectra (Figure 4c,d) reveals that the atomic ratio of Al/Zn in the ZnO lattice is 4.2% and 7.2% for AZO I and AZO IV, respectively. The result agrees well with the nominal molar ratio of Al nitrates of 5 mol% in the precursor solution, which shows an excellent doping efficiency of the NRs. In literature, the concentration of dopants in ZnO NRs is lower than the nominal values, because there is no warranty that the dopants ions added to the precursor’s solution will participate in the growth of NRs. Moreover, the results suggest an enhanced doping efficiency in alkaline media (pH = 11) compared to acidic/neutral media (pH = 5.5–6.5) of NRs growth.

### 3.2. Nanorod and Surface Chemistry

Figure 5a shows the XPS survey spectra for ZnO I, ZnO IV, AZO I, and AZO IV samples. The XPS analysis reveals the core-level, valence-level, and Auger peaks of Zn, O, and Al. In addition, other peaks corresponding to adventitious carbon (sub-layer contamination) are identified. The charge correction for the binding energies was employed in all spectra by adjusting the adventitious C1s, C-C peak to 284.8 eV. The high-intensity XPS peaks, namely the Zn2p, O1s, Al2p, and C1s, were used to compute the elemental and chemical composition (Table 3). This was done by quantifying the area of each peak, after subtracting a Shirley spectral background, and taking into account the relative sensitivity factor and escape depth correction. The results show that the Al/Zn ratio in AZO samples is about 6% closer to the nominal value in both AZO surfaces suggesting overall effective doping of the ZnO NRs. Moreover, the NRs surface seems richer in oxygen than zinc, and this trend is more pronounced in AZO surfaces. 

The high-resolution XPS spectra of the Zn2p, O1s, and Al2p, were also recorded. Figure 5b shows the two peaks corresponding to the spin-orbit splitting of Zn 2𝑝 3/2 and Zn 2𝑝 1/2 at binding energies of about 1021.75 and 1244.88 eV with an energy difference of 23.1 eV between the peaks. These results shift only slightly in the four samples and are found to match well reported data [50,51].

The high-resolution spectra of Al 2p reveal a peak with a wide range of binding energies ranging from 72 to 78 eV with peak values of about 74.4 eV, and with a noticeable intensity confirming the Al-doping of ZnO NRs. The reported binding energies for pure Al_2_O_3_ crystal range from 74.1 to 74.4 Ev [52], while those corresponding to aluminum hydroxide range from 74.8 to 75.2 eV [53]. This suggests the existence of oxides and hydroxides Al bonds at the surface of AZO samples.

The high-resolution O1s peaks in Figure 5d,e are all fitted by three components centered at 530.15 ± 0.15 eV (O_I_), 531.3 ± 0.3 eV (O_II_), and 532.5 ± 0.5 eV (O_III_). Generally, the component O_III_ is ascribed to the presence of chemisorbed oxygen as adsorbed water and carbides sub-layer [54]. The component O_I_ corresponds to the oxygen in lattice positions of ZnO, while the component O_II_ could be attributed to hydroxide and oxygen point defects [55]. The quantification of the components’ intensities (peak area), reveals that while the contribution of O_I_ is predominant on ZnO surfaces, O_II_ is the dominant contribution on AZO surfaces (Table 4). We note that the high oxygen content in AZO surfaces, computed from survey spectra, could be attributed to the high level of O_II_ component. Moreover, these findings suggest that Al doping promotes the formation of oxygen point defects at the surface of the nanorods.

### 3.3. Thermodynamic Simulation

Figure 6 shows the thermodynamic calculation of the solution under the experimental growth conditions (concentrations, temperature ~95 °C, and pH). Specifically, the figures show the pH-dependent speciation diagrams of Zn ions/complexes and Al ions/complexes present in the growth solution. It is worth mentioning that the speciation diagrams are simulated for the different experimental concentrations of ammonia used during the growth (See Table 1). Moreover, the region of interest of pH (highlighted in grey color) is based on the measurement of the pH of the solution before and after the growth of NRs. We also include the effect of temperature as the pH of the solution generally decreases while increasing the temperature.

For Zn(II) ions, the dominant species vary depending on the initial pH of the growth environment, which is adjusted by adding of NH_3_. The dominant Zn(II) species are all positively charged for all the considered ammonia concentrations. In neutral/acidic solutions, the dominant species is Zn^2+^ followed by some fractions of Zn(OH)^+^ or Zn(NH_3_)^2+^. In alkaline solutions, Zn^2+^(aq) ions also react with $NH_{4}^{-}$(aq) and form Zn(II) amine complexes mainly the dominant $Zn{\left(NH_{3}\right)}_{4}^{2+}$ complex. For Al(III) species, the dominant species, at low pH (5–6.5), is neutral Al(OH)_3_ followed by $Al{\left(OH\right)}_{2}^{+}$ and $Al{\left(OH\right)}_{4}^{-}$. The exact proportion of the negative and positive Al(III) species cannot be determined as it requires knowledge of the in situ pH during the growth. Under alkaline-pH conditions, negatively charged $Al{\left(OH\right)}_{4}^{-}$ ions were the only dominant Al(III) species.

### 3.4. Proposed Growth Mechanism

It is well known that supersaturation, the difference between solute concentration and solubility, is the driving force for crystal nucleation and growth in liquid environments. The Gibbs free energy of nucleation, for a spherical cluster, is given by:(2)$$∆G=\frac{4}{3}\pi r^{3}∆G_{v}-4\pi r^{2}\gamma$$
where *r* stands for the cluster radius, γ for the surface tension, and $∆G_{v}$ is free volume energy change. During the crystallization process, $∆G_{v}$ is related to the supersaturation by:(3)$$∆G_{v}=-\frac{kT}{\Omega }lnS$$
where $S=C/C_{0}$ is the supersaturation ratio. *C*, *C*_0_, and Ω are the concentration of solute, the solubility, and the atomic volume, respectively. The rate of nucleation depends non-linearly on supersaturation as follows [56]:(4)$$J=A\;S\;\exp\left(-\frac{B}{\ln^{2}S}\right)$$
with *A* and *B* constants. As a rule of thumb, at low supersaturation, the crystal grows faster but its nucleation is difficult resulting in large crystals. In contrast, high supersaturation promotes nucleation than growth, resulting in dense and small crystals [57,58,59]. Furthermore, increasing the supersaturation allows the transition from heterogeneous to homogenous nucleation/growth. It is worth mentioning that the growth of ZnO NRs on a seeded substrate, by hydrothermal process, is heterogeneous. Accordingly, controlling the supersaturation is a key parameter in tuning the morphology of the nanorods.

#### 3.4.1. Neutral/Acidic Growth Solution

For ZnO I sample, a classical growth mechanism [45,60] involving slow hydrolysis of HMTA producing OH^−^ anions can be used to explain the morphology of a well-aligned and high aspect ratio ZnO NR array. The zinc nitrates Zn(NO_3_)_2_ dissociate and provide Zn^2+^ ions, and HMTA acts as a pH buffer agent, which progressively releases OH^−^ ions following the pH-dependent gradual hydrolysis reactions: (5)$${\left(CH_{2}\right)}_{6}N_{4\left(aq\right)}+6H_{2}O_{\left(l\right)}\;⇌6HCHO_{\left(aq\right)}+NH_{3\left(aq\right)}$$
(6)$$NH_{3\left(aq\right)}+H_{2}O_{\left(l\right)}\;⇌NH_{4\left(aq\right)}^{+}+OH_{\left(aq\right)}^{-}$$

The Zn^2+^ ions and the OH^−^ ions react to first form the Zn(OH)_2_ complex and after dehydration, the ZnO crystal nuclei form. If the growth is carried out on a pre-seeded substrate, heterogeneous growth occurs, allowing the formation of ZnO NRs. It is also believed that the HMTA attaches to the non-polar faces of the ZnO crystal, promoting the growth along the *c*-axis and leading to high aspect ratio ZnO NRs [45].

Moreover, under pH of ~6.5, a high degree of supersaturation occurs and both homogeneous and heterogeneous growth takes place in the solution [49]. The Zn^2+^ ions available in solution, from dissolved zinc nitrates, will be used not only to grow ZnO NRs (heterogeneous) but also for precipitation (homogenous/heterogeneous) in solution. Accordingly, NRs with lower diameters are expected. Moreover, the nucleation rate is high leading to a dense and aligned NR array; since NRs growing in other directions will be suppressed by the neighboring aligned NRs, as shown in SEM micrographs (Figure 2e).

For the AZO I sample, the pH value of the solution at room temperature decreases and the growth solution becomes more acidic. This is due to aluminum and zinc hydroxide complexes forming in the solution and releasing *H*^+^ ions. Note that the in situ pH during the growth can be even more acidic due to the high temperature of 90 °C. As the pH becomes acidic (Table 1), the supersaturation decreases (solubility increases) leading to more heterogeneous growth rather than homogenous growth. In other words, more Zn^2+^ ions will be used to grow NRs rather than precipitation in the solution. Moreover, crystal growth is more favorable than nucleation, which leads to less dense and large AZO NRs as was evidenced by SEM images (Figure 3a). Due to their large diameter and low density, the NRs can grow in all directions leading to flower-shaped morphology. These results agree with Baruah and Dutta [44], Amin et al. [48], and Liu and Gao [49] reported ZnO large-diameter nanorods when prepared in acidic growth conditions.

#### 3.4.2. Alkaline Growth Solutions

For ZnO III and AZO III samples, the NRs morphology changed into a thin film-like structure with low aspect ratio NRs as revealed by SEM images (Figure 3e). The lowest solubility of ZnO occurs at this pH range of 9–9.5 [61] leading to very high supersaturation. This will increase homogenous growth (precipitation) rather than heterogeneous NRs growth. Indeed, the precursor solution at pH of 9.5 has a white/milky color confirming a precipitation process starting at room temperature. At other pH values, the prepared precursor solution is transparent which means absence or low precipitation. Moreover, high supersaturation at pH 9.5 promotes nucleation than crystal growth leading to dense and short NR array as evidenced by SEM imaging.

Although the supersaturation is high at pH values (10.5–11), SEM images show a well-aligned and high aspect ratio NR arrays for both ZnO and AZO samples. This is attributed to electrostatic interactions between charged species in solutions and the crystalline planes of the ZnO NR single crystals. Indeed, by using atomic force microscopy measurement [62], it has been shown that the surface charges of ZnO crystalline plans are pH-dependent in aqueous solutions. In neutral-acidic pH, the c-plan $\left(0001\right)$ and m-plan $\left(10\bar{1}0\right)$ are positively charged, whereas, in alkaline solutions, typically at pH = 11, the $\left(0001\right)$ is positively charged while $\left(10\bar{1}0\right)$ is negatively charged. This plays an important role in the growth mechanism, as charged ions may be selectively attracted to specific crystal plans by electrostatic interactions. The change in surface charges is attributed to the IEP, the isoelectric point ranging from 8 to 10 [63,64,65].

According to the thermodynamic simulation, in the pH range of 10–11, the predominant positively charged $Zn{\left(NH_{3}\right)}_{4}^{2+}$ ions will be attracted to the negatively charged $\left(0001\right)$ *c*-plane. Furthermore, the dominant $Al{\left(OH\right)}_{4}^{-}$ attracted to the positively charged m-planes $\left(10\bar{1}0\right)$ acts as a capping agent reducing the radial growth and promoting the growth along *c*-axis (Figure 7). This could explain why AZO NRs grown at pH = 11 are well aligned with a high aspect ratio. In addition, in alkaline media, the kinetic of growth along crystalline directions is different and found to be $V_{\langle 0001\rangle }>V_{\langle 10\bar{1}0\rangle }>V_{\langle 000\bar{1}\rangle }$. The kinetic growth along the *c*-axis is greater than the radial growth resulting in a very high aspect ratio of ZnO NRs. Moreover, (0001) plane having the highest crystal growth will be more etched than other planes resulting in a pen-like shape of the NRs tip. These findings are in agreement with Laudise et al. [66,67] where single crystals ZnO with similar shapes are produced by hydrothermal method under alkaline conditions (1–2 mol/L NaOH).

Another interesting result relates to the Al-doping mechanism of ZnO NRs under neutral/acidic and alkaline conditions. At high pH values, as mentioned above, the electrostatic interactions attract the dominant $Al{\left(OH\right)}_{4}^{-}$ species to the m-planes which could be the process by which the doping occurs, i.e the incorporation of Al ions into the ZnO crystal. At low pH (6.5–5), however, the dominant Al(III) species in the neutral Al(OH)_3_ followed by fractions of negatively charged $Al{\left(OH\right)}_{4}^{-}$ species. Although by the same process, the negative Al(III) ions could be attracted electrostatically to the positive planes, this cannot explain the high doping concertation of about 4.2% of Al into ZnO NRs as evidenced by TEM elemental maps. In addition to electrostatic interactions, doping can also occur by adsorption of all Al(III) species on ZnO NR crystals.

### 3.5. Nanostructure-Dependent Optical Properties

The optical properties of ZnO and AZO are expected to change by the nanostructures (induced by pH changes) and the Al-doping. Figure 8 shows the UV-Vis absorption spectra of all samples at various pH. The absorption edge is in the UV region at wavelengths slightly lower than 400 nm as expected from the wide-bandgap oxides (Eg > 3 eV). However, the shape of the spectra around the absorption edge is found to be greatly affected by the nanostructure. Indeed, for ZnO II sample (pHi = 6.55), a sharp increase of the absorption value around the absorption edge is observed. On the contrary, in the other ZnO samples, a significant steady increase of the absorption in the visible region is observed with an absorption peak of around 500 nm. This can be attributed to the change of the nanostructure from thin film/array (ZnO II) to aligned long NR array in other ZnO samples. We point out that the changes in the nanorod’s tip shape, from rod-like to pencil-like, do not affect the optical absorption. Similar features are observed in AZO samples, where an increase in the optical absorption in visible range in AZO I and IV samples that are characterized by long NR. In AZO II and III samples, with thin film/array structure, the absorption in the visible range is insignificant.

As the absorption in the visible range increases, the absorption edge seems red-shifted, which corresponds to a decrease in the energy band gap. Tauc’s plots, constructed from the absorption spectra, were used to estimate the band gap energy values (Table 5). At the absorption band edge of a direct semiconductor, like ZnO, the absorption coefficient α is related to the photon energy hv by the following formula:(7)$$\alpha \propto \frac{\sqrt{hv-E_{g}}}{hv}$$

Tauc’s plot consists of plotting (α hv)^2^ vs. hv; the linear part gives the values of the band gap energy [68]. The energy Eg values seem to decrease when the nanostructure changes from a thin film to a long-NR array. Indeed, ZnO-II has the highest bandgap energy for ZnO samples, which have a thin film-like structure. Similarly, the highest bandgap value is recorded for AZO II and III. Moreover, Al-doping tends to increase the bandgap energy as expected from the Burstein-Moss effect [69]. The exception of this rule is found in ZnO II and AZO II where a bandgap narrowing is observed upon doping. The Al-doping induces a formation of a trap level just below the conduction band edge; at higher doping concentration a new band is formed and a bandgap narrowing is expected. However, at very high doping concentration (degenerated semiconductor) the fermi-level is shifted up inside a conduction band leading to an optical bandgap widening. The combined effect of the nanostructure and Al-doping, inducing both variations of the optical bandgap and the enhancement of absorption in the visible range, play a crucial role in many applications such as photocatalytic properties [32,33].

The enhanced absorption in the visible range could be related to deep-level defects within the bandgap. We propose to explore the photoluminescence spectra to obtain information about the radiative recombination centers associated with crystal defects. Figure 9 shows the room-temperature photoluminescence spectra recorded on all ZnO and AZO samples where the excitation wavelength of 325 nm is used; energy of 3.8 eV well above the band energy gap. Multi-peak Gaussian fitting (with fit’s goodness of 0.99) is employed to fit the PL spectra and the PL spectra are found to exhibit 6 emission peaks within 350–600 nm wavelength range. Additional emission peaks, not fitted, are found resulting in red emission 700–900 nm for all samples. It is important to emphasize that the origin of the emissions in ZnO is highly controversial in literature, probably because many deep-level defect centers co-exist within ZnO bandgap [70]. The less controversial one is the UV-emission peak (labeled P1) that corresponds to the band-to-band emission where an electron at the exciton states FX (located at 0.06 eV below CB) recombines with a hole from the VB. This emission occurs at about 394–406 nm which matches well the optical bandgap measured with UV-Vis spectroscopy. The violet, violet-blue, and blue emissions result in 4 emission peaks at about 420 nm, 440 nm, 460 nm, and 485 nm, respectively. These emissions can be attributed to the zinc defects involving zinc interstitials Zn_i_ and Zinc vacancies V_Zn_ energy levels. Zn_i_ level is 0.22 eV below the CB, V_Zn_ level is located at 0.3 eV above the VB, and additional extended Zn_i_ peaks (ex-Zn_i_) corresponding to charged Zn_i_^+^ and Zn_i_^++^ defect states are located at 0.54 and 0.63 eV below the CB; values obtained for ZnO with bandgap of 3.36 eV [70,71,72]. Assuming these values and considering the estimated bandgap energies (Table 5), the following conclusions are made. The violet emission at 420 nm (Peak 2) can be attributed to either a transition (Zn_i_ → VB) ~418 nm or (CB → V_Zn_) ~427 nm. The emission peaks 3 and 4 correspond to an electron captured at the extended ex-Zn_i_ recombining with a hole from VB, as several authors associated these transitions with emission at 440–460 nm [71,72,73,74]. Another possibility could be a recombination of an electron from Zn_i_ level with a hole from the V_Zn_ level, which results in an emission of about 460 nm (Peak 3). The fourth emission peak at ~485 nm could be attributed to the recombination of an electron from ex-Zn_i_ level with a hole from V_Zn_ level as reported by R. Khokhra et al. [74]. It is worth mentioning that according to ab-initio calculation, Zn_i_ defects have high formation energy and is difficult to form under equilibrium conditions. However, the fast NR growth by microwave-assisted hydrothermal method, considered non-equilibrium conditions, may lead to the formation of such defects. Moreover, for AZO samples, the Al has smaller ionic radii than Zn, Al substituting Zn results in larger interstitial sites which favor the formation of Zn_i_ defects.

The green emission falls in 500–560nm spectral region and peaks at 527 nm (2.35 eV) for all samples. Although its origin is still debated, most authors attribute it to a singly charged vacancy oxygen V_o_^+^ defect [71,72,74,75,76,77]. However, the Gaussian fitting reveals a board emission peak which suggests the existence of other transitions resulting in green emission. The V_o_^+^ defect, being unstable, could form a neutral state V_o_ by the capture of an electron from the CB. It could also form a V_o_^++^ state by capturing a hole from the VB. V_o_ located at 0.86 eV below the CB may result in a green emission at 500 nm, whereas V_o_^++^, located at 1.16 eV above the VB, may results in an emission at 560 nm [71,74]. Others reported contributions to green emissions including Zn_i_, V_Zn_, and oxygen interstitials O_i_ [78,79,80]. The PL spectra also reveals the existence of red emission with a weak intensity peak ranging from 750–900 nm and centered at about 780–800 nm (~1.6 eV). This emission is attributed to transition involving deep-level states namely oxygen defects. A. Galdamez et al. [70] reported that V_o_ located at ~1.65 eV below CB results in an emission at 750 nm (CB → V_o_); O_i_ located at ~1.59 eV above VB, results in an emission at 780 nm (O_i_ → VB). V. Kumar et al. [81] located the V_o_ at ~1.85 eV below the CB resulting in a red emission of 760 nm (Zn_i_ → V_o_). M. Bedrouni et al. [73] reported on PL emission of ZnO thin film and showed attributed the red emission to the following transitions CB → V_o_, V_o_ → VB or H_i_ → V_o_. It is important to mention that the existence of oxygen vacancies is supported by XPS findings where a signature of these defects is found after deconvolution of the high-resolution oxygen peak.

The position (energy) of the emission peaks seems independent on the pH growth values (Table 6). The most interesting result, however, is that the relative intensity of the violet and violet-blue emission peaks to the UV peak depends strongly on the pH. As shown in Figure 9a,b, a redshift of the PL spectra with pH is observed which cannot be explained by estimated energy bandgap values (Table 5). This shift towards a higher wavelength is accompanied by a decrease in the relative emission intensity of the UV peak (P1). There is a clear transition of the main radiative emission from the UV (band-to-band) into violet or violet/blue emissions involving deeper defect levels (Zn_i_ and V_Zn_). For ZnO I, grown at neutral pH, the intensity of peak 1 and peak 2 are comparable with a ratio I_2_/I_1_ ~1.1; however, at basic pH, peak 3 has the dominant radiative emission and the ratio I_3_/I_1_ increases to reach ~2.6 for ZnO IV (Figure 10b). A similar tendency is observed for AZO samples where the relative intensity between the defect state to UV peak increases from ~1.3 to ~1.6. In summary, there is a gradual transition of the main radiative emission from the UV (band-to-band), to violet emission involving Zn_i_ level (0.22 eV below CB), and toward even a deeper level involving either V_Zn_ (0.3 eV above CB) and/or extended Zn_i_ levels (0.5–0.6 eV below CB). Our results suggest that growing ZnO or AZO NRs in alkaline solutions, favors the non-radiative transitions (i.e phonon relaxation) from CB to deeper Zni and ex-Zni levels. This corroborates with other results suggesting the Zni complexes as the main candidate for non-radiative recombination centers within ZnO NRs [82].

## 4. Conclusions

The well-aligned, dense NR arrays of pure ZnO and heavily doped AZO have been successfully grown by using a novel low-temperature microwave-assisted hydrothermal deposition technique. All samples exhibit single-crystal XRD patterns with reflections belonging to pure wurtzite ZnO crystalline structure and strong diffraction peak along (002) reflection. While the pH is found to strongly affect the intensity of the main diffraction peak, the Al-doping is found to induce new XRD reflections. FESEM scans reveal that pH and Al-doping strongly affect the morphology of the NR array in terms of nanorods’ aspect ratio, tip shape, and vertical alignment. While all ZnO samples grow into a well-aligned NR array, only the AZO sample, grown in alkaline media (pH = 11), is found to exhibit a good NRs alignment. The thermodynamic-based simulations of the growth solution have been performed to determine the pH-dependent speciation diagram under experimental conditions. The key parameters controlling the growth are found to be supersaturation and pH-dependent electrostatic interactions with charged m-planes and c-planes of ZnO NRs single crystals. The proposed mechanism included the combined effect of doping and pH on the NRs growth and morphology. The TEM elemental mapping of the spatial distribution of Al ions reveals that Al-doping appears to be homogenous and continuous over the NRs. XPS analysis reveals that Al-doping of ZnO NRS induces the formation of oxygen point defects that could be either oxygen vacancies or interstitials. These oxygen point defects could have strong implications on the optoelectronic and photocatalytic properties of ZnO NRs. The optical absorption spectra are found to be highly dependent on the pH value, and an enhancement of the visible absorption is evident. The fitting of the PL spectra reveals a shift of the main recombination from the UV (band-to-band) peak toward violet or violet/blue emission peaks involving Zn interstitials deep-defect levels. This shift and ratio between the emissions can be tuned by controlling the pH value. The synthesis of highly doped and well-aligned ZnO and AZO nanorods using this simple inexpensive method and the pH-dependent optical properties would increase the potential use of these nanomaterials in a wide variety of applications.

## Figures and Tables

**Figure 1 nanomaterials-12-03735-f001:**
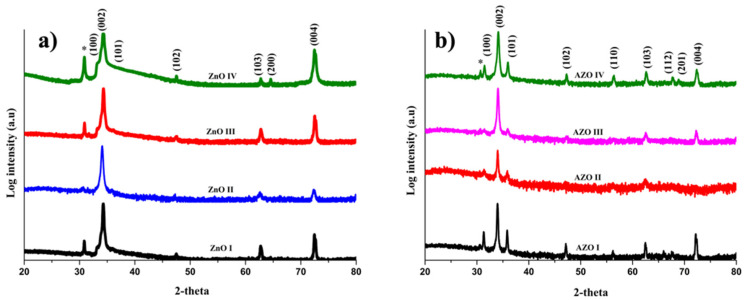
XRD pattern for: (**a**) ZnO and (**b**) AZO NRs grown at various pH. Peak (*) corresponds to Zn(OH)_2_ phase.

**Figure 2 nanomaterials-12-03735-f002:**
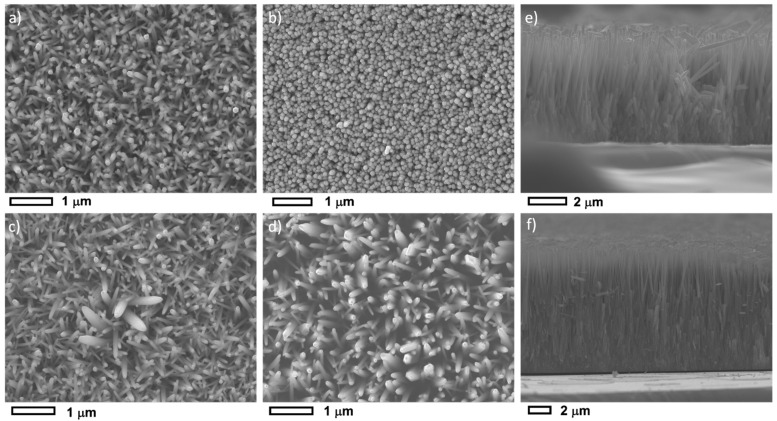
Top view FESEM images for ZnO NRs sample: (**a**) ZnO I, (**b**) ZnO II, (**c**) ZnO III, (**d**) ZnO IV. Top view FESEM images for (**e**) ZnO I, and (**f**) ZnO IV samples.

**Figure 3 nanomaterials-12-03735-f003:**
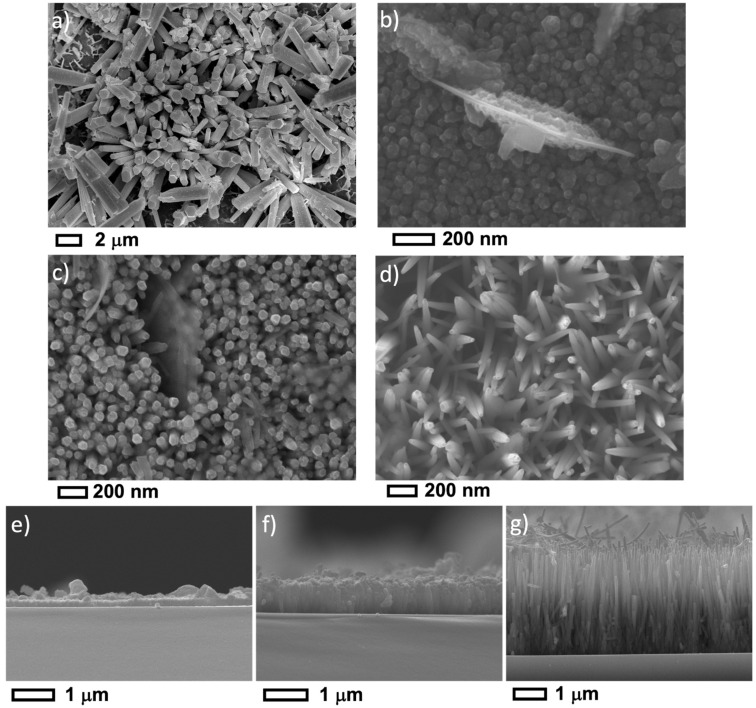
Top view FESEM images for AZO NRs samples: (**a**) AZO I, (**b**) AZO II, (**c**) AZO III, (**d**) AZO IV. Cross-section FESEM images for AZO NRs samples: (**e**) AZO II, (**f**) AZO III, and (**g**) AZO IV.

**Figure 4 nanomaterials-12-03735-f004:**
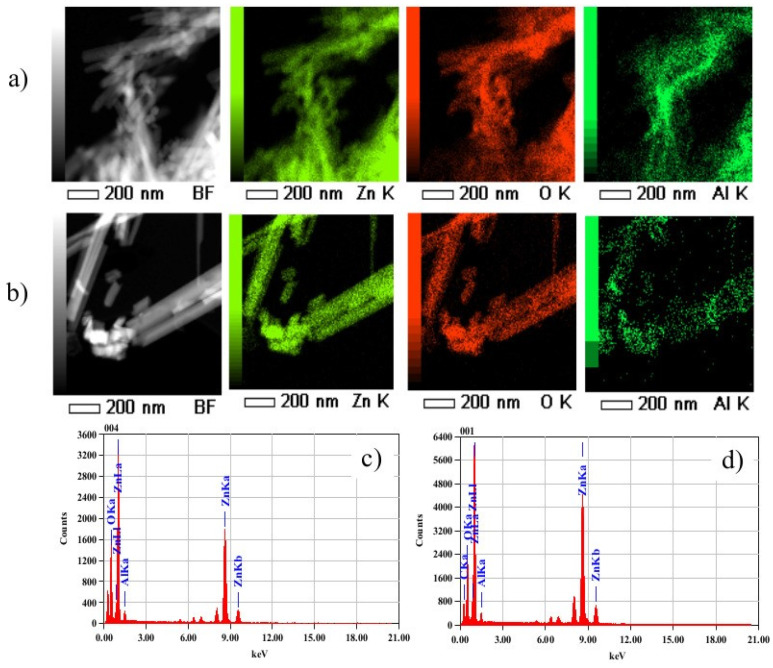
TEM scans and elemental mapping on AZO NRs samples: (**a**) AZO I, (**b**) AZO IV, and the corresponding EDX spectra showing (**c**) AZO I, and (**d**) AZO IV.

**Figure 5 nanomaterials-12-03735-f005:**
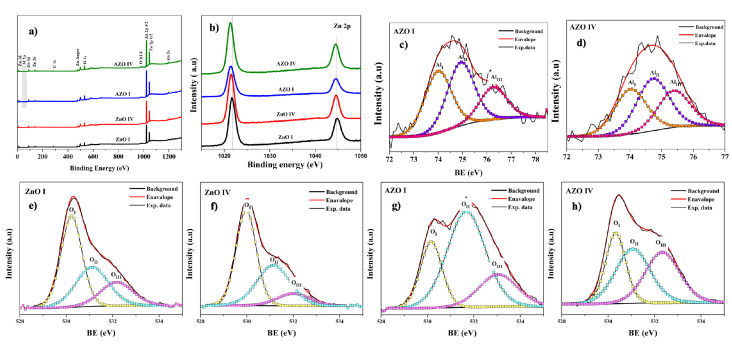
(**a**) XPS survey spectra for ZnO I, ZnO IV, AZO I, and AZO IV surfaces. High resolution of spectra of (**b**) Zn 2p, (**c**,**d**) Al2p, and (**e****–h**) O1s XPS peaks.

**Figure 6 nanomaterials-12-03735-f006:**
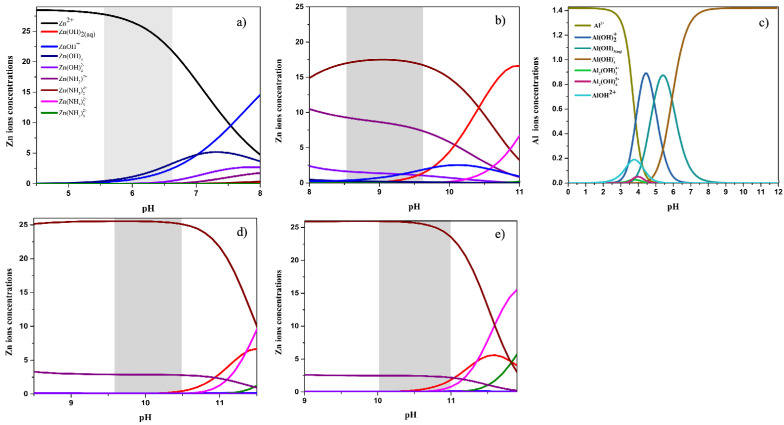
Theoretical calculations of ions concentrations as a function of pH using the Visual MINTEQ software: (**a**,**b**,**d**,**e**) Zn(II) speciation diagram at various pH and ammonia concentration, (**c**) Al(III) speciation diagram. Note that the HMTA is replaced by an equivalent amount of NH_3_.

**Figure 7 nanomaterials-12-03735-f007:**
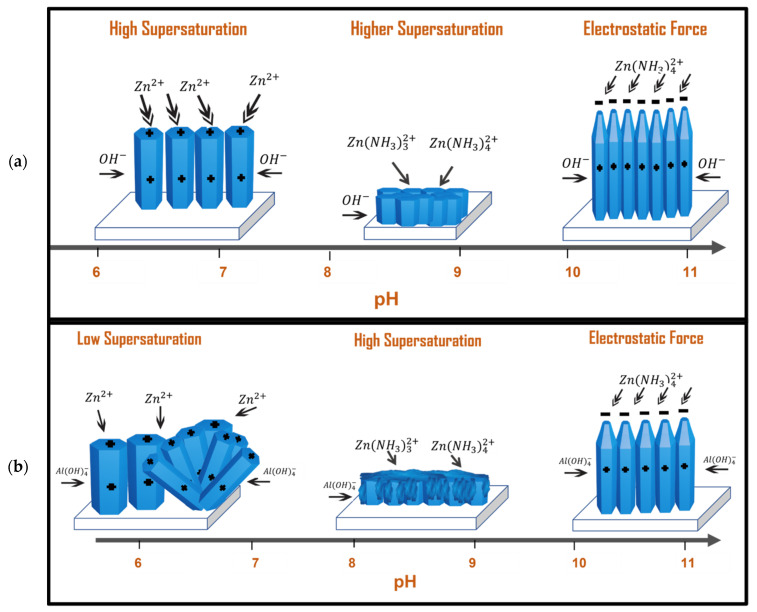
A schematic diagram that represents the growth mechanism of: (**a**) ZnO and (**b**) AZO NRs, in different pH precursors using microwave-assisted hydrothermal method.

**Figure 8 nanomaterials-12-03735-f008:**
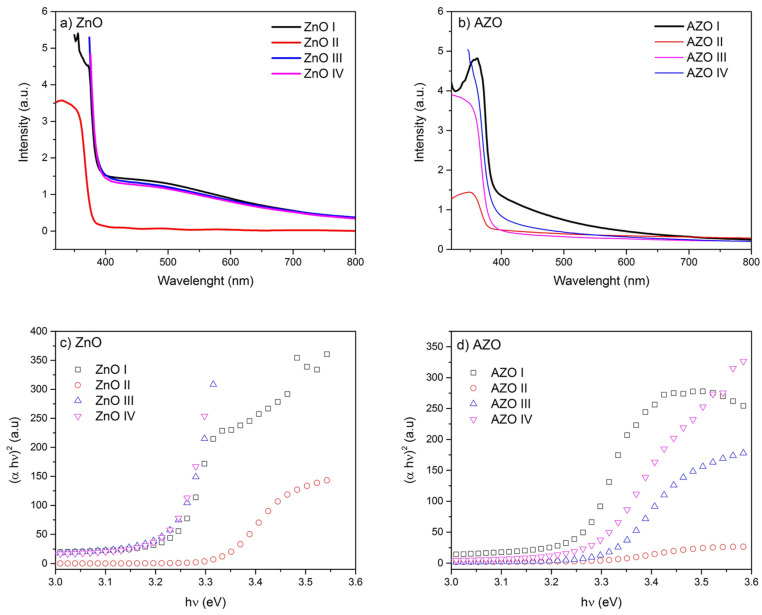
(**a**,**b**) UV-Vis absorption spectra and (**c**,**d**) the corresponding Tauc plots of ZnO and AZO samples.

**Figure 9 nanomaterials-12-03735-f009:**
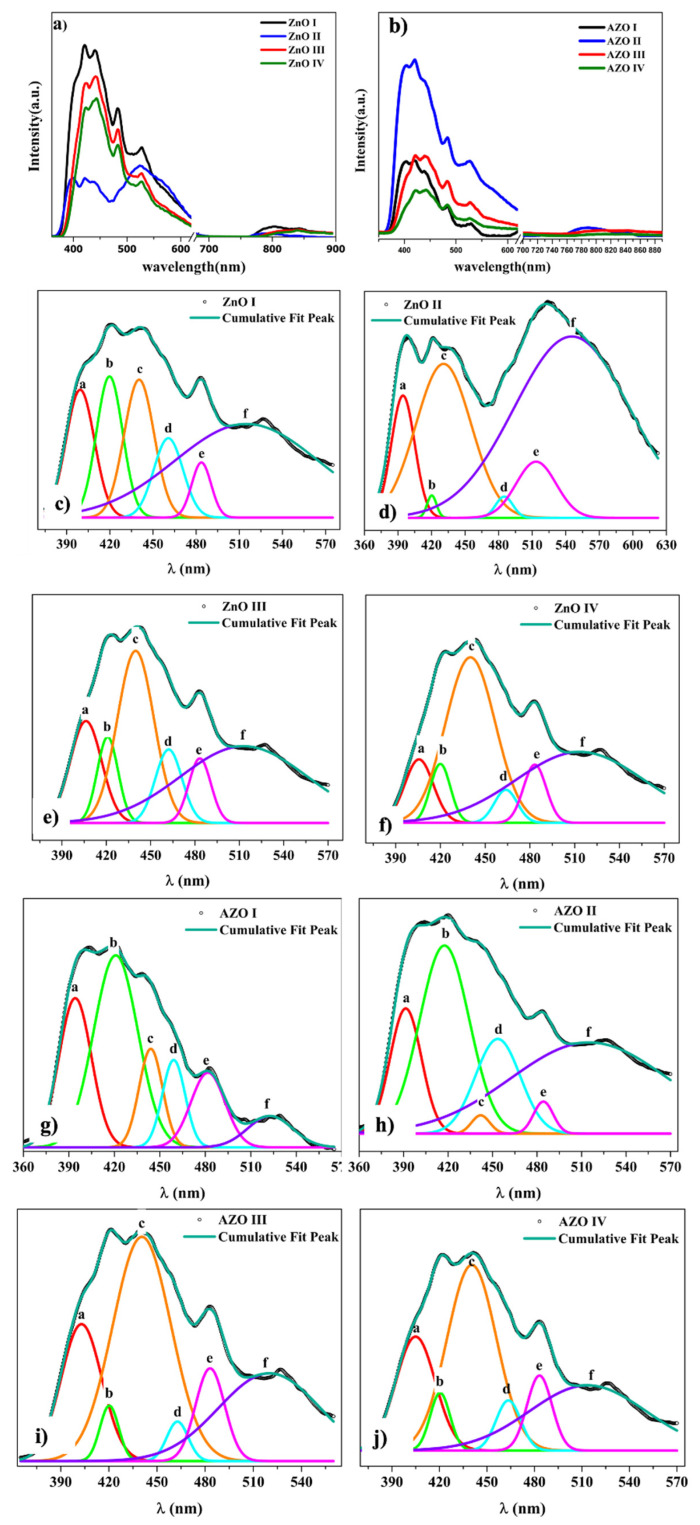
(**a**,**b**) Room temperature PL spectra and (**c**–**j**) PL deconvolution in the visible range of nanostructured ZnO and AZO samples.

**Figure 10 nanomaterials-12-03735-f010:**
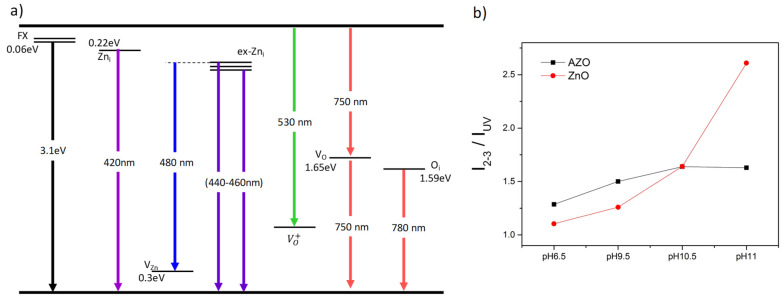
(**a**) Schematic diagram of the main PL radiative recombination centers, (**b**) the PL intensity ratio between the band-to-band emission I_UV_ (~395 nm) and the violet I_2_ (~420 nm) and blue-violet I_3_ (~430 nm) emissions.

**Table 1 nanomaterials-12-03735-t001:** Conditions used during the experiments to prepare ZnO and AZO NRs grew on glass substrates (pH_i_: initial pH, pH_f_: final pH after growth, NH_3_ volume added initially for specific pH, Concentration of NH_3_ resulted from HMTA and added NH_3_).

Sample ID	pH_i_	pH_f_	NH_3_ vol. (mL)	NH_3_ conc. (mM)
ZnO I	6.68	6.46	0.00	30.0
ZnO II	9.55	8.52	2.00	161.5
ZnO III	10.50	9.29	8.31	559.8
ZnO IV	11.00	9.67	15.05	959.4
AZO I	5.56	6.56	0.00	30.0
AZO II	9.55	8.77	2.50	194.0
AZO III	10.50	9.65	4.20	303.1
AZO IV	11.00	9.95	8.60	577.5

**Table 2 nanomaterials-12-03735-t002:** Crystalline properties of ZnO and AZO samples from XRD analysis.

Sample ID	2θ (°)	*D* (nm)	*c* (Å)	*a* (Å)	*c*/*a*	FWHM (°)	Bond Length (Å)
ZnO I	34.262	54	5.230	3.236	1.616	0.161	1.975
ZnO II	34.132	53	5.249	3.223	1.626	0.194	1.974
ZnO III	34.252	45	5.232	3.234	1.618	0.161	1.974
ZnO IV	34.194	66	5.240	3.219	1.628	0.132	1.970
AZO I	34.166	64	5.244	3.227	1.625	0.135	1.973
AZO II	34.031	42	5.265	3.228	1.631	0.205	1.976
AZO III	34.170	48	5.244	3.218	1.629	0.181	1.969
AZO IV	34.168	47	5.244	3.225	1.626	0.185	1.972

**Table 3 nanomaterials-12-03735-t003:** Elemental analysis of ZnO and AZO NRs array surface extracted from XPS spectra. The content of each element is given in at %.

Sample ID	Zn	O	C	Al	O/Zn	Al/Zn
ZnO I	39.4	45.3	15.4	-	1.15	-
ZnO IV	42.2	46.2	11.6	-	1.09	-
AZO I	29.6	52.4	16.0	2	1.77	6.7%
AZO IV	37.0	49.2	11.4	2.4	1.33	6.5%

**Table 4 nanomaterials-12-03735-t004:** Results of deconvolution of high-resolution spectra of O1s peak.

Sample	O_I_		O_II_		O_III_	
	BE (eV)	Area %	BE (eV)	Area %	BE (eV)	Area %
ZnO I	530.2	48.2	531.1	32.3	532.2	19.5
ZnO IV	530.0	52.5	531.1	36.5	532.0	11.0
AZO I	530.2	22.7	531.7	57.8	533.0	19.5
AZO IV	530.3	28.1	531.1	37.3	532.3	34.6

**Table 5 nanomaterials-12-03735-t005:** The energy bandgap values for the respective samples.

Sample ID	Eg (eV)
ZnO I	3.23
ZnO II	3.34
ZnO III	3.24
ZnO IV	3.28
AZO I	3.26
AZO II	3.29
AZO III	3.31
AZO IV	3.28

**Table 6 nanomaterials-12-03735-t006:** PL deconvolution results on ZnO and AZO samples.

Sample	Peak Position (nm)	FWHM (nm)	Area (a.u)	Averaged Area (%)
ZnO I	399 420 440 461 484 529	23 21 24 24 16 117	397 405 452 254 117 1471	12.8 13.1 14.6 8.2 3.8 47.5
ZnO II	395 420 431 485 513 545	25 10 58 18 42 121	148 11 439 19 116 1080	8.1 0.6 24.2 1 6.4 59.6
ZnO III	406 421 440 462 483 527	25 17 29 21 18 101	299 167 587 183 134 906	13.1 7.3 25.8 8 5.9 39.8
ZnO IV	406 420 440 464 483 527	22 16 39 18 18 101	154 100 440 64 112 768	9.4 6.1 26.9 3.9 6.8 46.9
AZO I	394 421 444 459 482 529	25 34 18 18 26 34	397 694 189 163 209 114	22.5 39.3 10.7 9.2 11.8 6.5
AZO II	392 418 442 453 484 527	25 39 14 35 16 125	724 1718 62 784 116 2640	12 28.4 1 13 1.9 43.7
AZO III	403 420 441 463 483 527	29 14 43 16 21 75	390 73 914 59 189 623	17.3 3.2 40.7 2.6 8.4 27.7
AZO IV	405 420 440 463 483 527	29 15 36 18 19 86	207 53 422 57 87 350	17.6 4.5 35.9 4.9 7.4 29.8

## Data Availability

Not applicable.

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
