# Peer review of "pH Controlled Nanostructure and Optical Properties of ZnO and Al-Doped ZnO Nanorod Arrays Grown by Microwave-Assisted Hydrothermal Method"

_nanomaterials, 2022, doi:10.3390/nano12213735_

Round 1
Reviewer 1 Report
...
Manuscript titled "pH-controlled nanostructure and optical properties of ZnO and AZO nanorods arrays grown by microwave-assisted hydrothermal method" by Lamia Al-Farsi, T.M. Souier, Muna Al Hinai, M. T. Zar Myint, H. Htet Kyaw, H.M. Widatallah, M. Al-Abri is devoted to studying the effect of synthesis conditions on the physicochemical properties of zinc oxide nanorods and aluminum-doped zinc oxide. The content of the manuscript as a whole corresponds to the subject of Journal Nanomaterials and can be published in it. But first I ask you to make some corrections and additions to the manuscript. (Please do not enter into correspondence with the reviewer. Make changes immediately to the manuscript.)
3 - Remove word hyphenation in the title of the manuscript.
45 - Please spell out the abbreviation (NR) at the first mention.
83, 84, 85, 86, 86 - Remove the comma before the "more" symbol.
86, "[(NH3),> 25%]" - If earlier such a record obviously meant the "purity" of the reagent, then in this case we are talking about the concentration of an aqueous solution. Consider changing the entry in a way that does not confuse the reader.
90 - "glass substrates" - Specify the glass composition. Describe its properties. Is it subject to hydrolysis under synthesis conditions? How did you check it?
99 "microwave oven under a power of 180W for 45 minutes." What temperature did the solutions reach under these conditions? How did the authors deal with the removal of ammonia from solutions when heated? Was an autoclave used? If yes, please provide a detailed description.
108 - There are no data in the table for samples "ZnO II" and "AZO II" which are mentioned below.
109 - Insert a space.
170 "showed a slight shift towards a lower angle" - This statement is inconsistent with the data in Figure 1, where the reflex (002) is around 44 degrees.
177 - "the Zn-O bond lengths" - Describe how this value was determined.
182 - The figure does not contain data for samples of the "III" series, which are mentioned in other figures and tables.
184 - The table does not contain data for samples of the "V" series, which are mentioned in other figures and tables. Indicate the errors in determining the quantities. The positions of the (002) reflections indicated for the samples of the "AZO" series do not correspond to the data in Figure 1.
184 - "cry. size (nm)" - Decipher the abbreviation.
233, 439 - "pHi" What does the symbol "i" mean?
275 - Specify the units of measurements. Check and report measurement errors.
300 - Figure caption and figure should be on the same page.
302 - What is the role of HMTA in synthesis? Does it affect pH? Why wasn't it used in thermodynamic simulations?
323 - For what temperature are these diagrams built? How is this temperature related to the synthesis temperature? (I believe that the pH of the solutions was measured at room temperature, and the synthesis was carried out at a very high temperature.) How do the conditions in diagrams a-d differ from each other? It seems that diagrams b and c are the same. This is true?
326 - The purpose of this section is not clear to the reviewer. These well-known expressions are not used in any way and are not discussed in the manuscript.
354 - "decompose" - Dissociate!
362 - "long-chain polymer," - Polymer? https://en.wikipedia.org/wiki/Polymer
392, 461 - ";" → "."
433 - Start your title with a capital letter.
437 - "lower than 400 nm as expected from the wide-bandgap oxides" - What does the spectrum of the glass substrate look like?
438 - "is found to be greatly affected by the nanostructure." - Is there any connection between the type of spectrum and the thickness of the coating?
450 - "Tauc's plots," - Give an explanation or reference to a literary source.
470 - No data on samples of the "V" series. Specify the error in determining the values.
472 - No data available for "V" series specimens.
Give a description of figures c and d in the caption.
473 - "could be related to the existence of deep-level defects within the bandgap" - How did the authors take into account the different thicknesses of the samples?
551 - The figure does not show data for samples of the "III" series.
552 - No data available for "V" series specimens. Check the errors in the values presented.
560 - Conclusions are too voluminous. Formulate brief conclusions from your work at the end of the manuscript. You don't need to list the results. Align the conclusions with the purpose of the work.
...
Author Response
The authors are grateful to the reviewer for his valuable comments and corrections that will enhance the quality of the paper. As per the editorial office's recommendation, the changes in the manuscript are marked up using the “Track Changes” function in MS/Word. In the attached cover letter there are the point-by-point answers to the reviewer's comments and corrections.

Reviewer 2 Report
Dear Authors, in your manuscript, the following points should be added/changed to further improve it:
1. Introduction: I have a comment on the sentence “ZnO possess the richest family of nanostructures among all materials; this includes 29 nanocombs, nanorings, nanocages, nanosaws, nanospirals, nanosprings, nanobelts, nan-30 owires, and nanorods [1-11].” The morphology of ZnO nanomaterials is extensively described in a review article (DOI:10.3390/nano10061086).
2. Introduction: Please add information that zinc oxide is a multifunctional material.
3. Introduction: The authors report the results of microwave synthesis of ZnO so please add one/two sentences about the advantages of the synthesis method used. I suggest pointing to review articles where readers can find a comprehensive description of microwave synthesis of nano ZnO (DOI:10.3390/nano10061086, DOI:10.1080/10408436.2017.1397501).
4. Introduction: Please introduce and explain the abbreviation "AZO" in the text.
5. Synthesis: Please add information about the microwave oven used (model, manufacturer).
6. Synthesis: Please provide information about Mettler Toledo meter (model).
7. Synthesis: I have a comment on the sentence “The 103 AZO NRs were prepared as in the previous steps except that aluminum nitrate nonahy-104 drate was added to the zinc nitrate hexahydrate precursor (Al/Zn molar ratio = 5 mol%) 105 while maintaining the precursors’ concentration at 30mM.” Please provide information on the Al doping content of ZnO in the samples you received.
8. Morphological and Structural properties: Please move the description of the Sherrer equation and the equation for calculating lattice parameters to the "Experimental details" section.
9. Morphological and Structural properties: I have a comment on the sentence “Figure1a shows XRD patterns for undoped ZnO NRs. All NRs arrays crystallize 138 within the hexagonal wurtzite crystalline structure in agreement with JCPDS card No. 04-139 009-7657 of the Zincite ZnO phase.” Please add the source (citation) for the data “JCPDS card No. 04-139 009-7657” for the reader to verify.
10. Morphological and Structural properties: I have a comment on the sentence “It should be noted that Al secondary phases, such as Al2O3, have not been observed.” The Al2O3 phase could not be seen because too low a synthesis temperature was used for the Al2O3 crystalline phase to exist . The authors should check whether the crystalline phase of Al(OH)3 is present in the samples obtained. In the discussion, it should be noted that the XRD method has a detection threshold for the foreign phase and even this can be as high as 5%mol depending on the sample composition. XRD analysis is unable to detect the presence of a foreign phase in the form of an amorphous phase.
11. Morphological and Structural properties: I have a comment on the sentences “However, Al-doping is found to 145 induce new crystalline planes namely (110), (112), and (201). Interestingly, another low 146 angle diffraction peak at about 2?~ 30.8° is also evidenced in some ZnO and AZO sam-147 ples mainly those grown in alkaline media (pH = 10.5 or 11).”
The XRD results for the AZO NRs samples are completely contradictory to their discussion. First, the XRD results contradict the receipt of the ZnO phase in the AZO NRs samples. Please compare the AZO NRs XRD results with the XRD result for the ZnO reference sample (Standard JCPDS card number 36-1451, DOI:10.1155/2016/2789871). Second, there is no trace of crystalline planes (110), (112), and (201) on the XRD results of AZO NRs. Once again, I encourage the authors to compare the XRD results with Standard JCPDS card number 36-1451 for verification. Third, the XRD results for the AZO NRs samples show no diffraction peaks at an angle of 2?~ 30.8°.
Related to the evidence (Figure 1, XRD AZO NRs), I conclude that there was no ZnO crystalline phase in any sample of AZO NRs. The authors did not obtain Al-doped ZnO NRs as claimed. Therefore, I believe that the manuscript should be rejected.
Author Response

(The authors gave the same response as above.)

Reviewer 3 Report
This manuscript's work entitled “pH-controlled nanostructure and optical properties of ZnO and AZO nanorods arrays grown by microwave-assisted hydrothermal method” is interesting and not well presented, with multiple technical flaws found regarding the importance of doping materials and their applications, etc. The article has many grammatical and sentence errors, and the language organization needs to be improved. For these reasons, I conclude that the paper should undergo major revision
1. Text formatting is not as per journal parts.
2. Format the text throughout the manuscript and remove hyperlinks in section 2.3. Characterization’ and lines No: 122-124.
3. Line 148-149: Provide a proper reference for the following phase “A similar peak is observed 148 by some authors and is attributed to zinc hydroxide Zn(OH)2 phase.”
4. Perform the deconvolution of XPS spectra data and provide the gaussian value of XPS spectra to understand the chemical composition in the form of a table
5. There are many grammatical and sentence errors in the article, and the language organization needs to be improved.
6. Authors may provide more significant details about the importance of doping in conclusion on the future scopes and challenges with possible applications.
7. Reference formatting is found as per the journal. Verify and correct.
Author Response

(The authors gave the same response as above.)

Round 2
Reviewer 1 Report
The authors accepted and corrected virtually all of the reviewer's criticisms, so I can only support the publication of this manuscript.
Author Response
Response to Reviewer 1 Comments
Point 1: The authors accepted and corrected virtually all of the reviewer's criticisms, so I can only support the publication of this manuscript.
Response 1: The authors are grateful to the reviewer for his thoughtful comments and efforts towards improving the quality of the manuscript.
Reviewer 2 Report
1. Characterization: I have a comment on the sentence “High-resolution transmission electron microscope 144 (HRTEM) by a JEOL (JEM-2100F) with an accelerating potential of 200 kV was used to 145 perform elemental mapping on AZO NRs along withand energy-dispersive X-ray spec-146 troscopy EDS.” Please add information about the EDS analyzer used (model, manufacturer).
2. Figure 2 and Figure 3: Please significantly improve the readability of the scale bar on all SEM images.
Author Response
Response to Reviewer 2 Comments
The authors are grateful to the reviewer for his thoughtful comments and efforts towards improving the quality of the manuscript. As per the editorial office's recommendation, the changes in the manuscript are marked up using the “Track Changes” function in MS/Word. Hereafter, are the point-by-point answers to the reviewer's comments and corrections.
Point 1: Characterization: I have a comment on the sentence “High-resolution transmission electron microscope 144 (HRTEM) by a JEOL (JEM-2100F) with an accelerating potential of 200 kV was used to 145 perform elemental mapping on AZO NRs and energy-dispersive X-ray spec-146 troscopy EDS.” Please add information about the EDS analyzer used (model, manufacturer).
Response 1: We thank the reviewer for the comment. As per the suggestion, the corresponding text is edited as follow:
"High-resolution transmission electron microscope (HRTEM) by a JEOL (JEM-2100F), operating at an accelerating potential of 200 kV and equipped with energy-dispersive X-ray spectrometer (EDS, Joel EX-24063), was used to perform elemental mapping on AZO NRs and the corresponding EDS spectra."
Point 2: Figure 2 and Figure 3: Please significantly improve the readability of the scale bar on all SEM images.
Response 2: We agree with the reviewer comment. As per the recommendation, the figures 2 and 3 are improved (SEM images with improved scale bar)